# The Impact of Hepatitis C Virus, Metabolic Disturbance, and Unhealthy Behavior on Chronic Kidney Disease: A Secondary Cross-Sectional Analysis

**DOI:** 10.3390/ijerph19063558

**Published:** 2022-03-17

**Authors:** Po-Chang Wang, Yi-Fang Wu, Ming-Shyan Lin, Chun-Liang Lin, Ming-Ling Chang, Shih-Tai Chang, Tzu-Chieh Weng, Mei-Yen Chen

**Affiliations:** 1Division of Internal Medicine, Department of Cardiology, Chang Gung Memorial Hospital, Chiayi 613, Taiwan; myheart88168@gmail.com (P.-C.W.); mingshyan@gmail.com (M.-S.L.); cst1234567@yahoo.com.tw (S.-T.C.); claire2080@hotmail.com (T.-C.W.); 2Department of Emergency Medicine, Chang Gung Memorial Hospital, Chiayi 613, Taiwan; yvonnearea@gmail.com; 3Graduate Institute of Clinical Medical Sciences, College of Medicine, Chang Gung University, Taoyuan 333, Taiwan; 4Department of Nephrology, Chang Gung Memorial Hospital, Chiayi 613, Taiwan; linchunliang@adm.cgmh.org.tw; 5Kidney Research Center, Chang Gung Memorial Hospital, Taoyuan 333, Taiwan; 6College of Medicine, Chang Gung University, Taiyuan 333, Taiwan; mlchang8210@gmail.com; 7Division of Hepatology, Department of Gastroenterology and Hepatology, Chang Gung Memorial Hospital, Taoyuan 333, Taiwan; 8Department of Nursing, Chang Gung University of Science and Technology, Chiayi 613, Taiwan; 9School of Nursing, Chang Gung University, Taoyuan 333, Taiwan

**Keywords:** chronic kidney disease, hepatitis C virus, metabolic syndrome, exercise, dietary habits

## Abstract

Background: Hepatitis C virus (HCV) infection is associated with a higher risk of chronic kidney disease (CKD). This study investigates the relationship among HCV, CKD, and understudied confounders, such as unhealthy behaviors and metabolic disturbances. Methods: This cross-sectional study was conducted as part of a community health promotion program in an HCV endemic area of Taiwan from June to December 2019. Multivariable logistic regression analyses adjusted for demographic and clinical characteristics were performed to investigate the association between CKD and HCV seropositivity. Results: Of 2387 participants who underwent health check-ups, the mean age was 64.1 years old; females predominated (63.2%), and 306 (12.8%) subjects were seropositive for HCV. CKD, defined as a lower estimated glomerular filtration rate (eGFR) was associated with unhealthy dietary habits, metabolic syndrome, and HCV. Less frequent exercise, higher waist circumference (WC) and HbA1c all affected risk of CKD; HCV increased risk of CKD by 44% compared to non-HCV (OR 1.44, 95% confidence interval (CI) 1.05–1.98) in the multivariable analysis. In the HCV group, lower eGFR was also significantly associated with the severity of metabolic syndrome (MetS) (median eGFR was 86.4, 77.1, and 64.5 mL/min/1.73 m^2^ for individuals with three and five MetS components, respectively). Conclusions: Beyond metabolic disturbance and irregular exercise, HCV seropositivity is independently associated with CKD in a community survey. Healthy lifestyle promotion might protect against renal function decline in HCV; however, the mechanisms underlying the association need further large-scale investigation.

## 1. Introduction

Hepatitis C virus (HCV) infection affects approximately 71.1 million people worldwide with a global prevalence of 1–2% [1], and Southern Taiwan is an HCV endemic area with a prevalence higher than 4% [2]. Chronic HCV infection leads to multiple hepatic complications, including advanced liver cirrhosis, hepatocellular carcinoma, and worsening renal function [3,4,5], leading to a twofold higher rate of all-cause mortality [6]. Chronic HCV patients have a higher risk of developing membranoproliferative glomerulo-nephritis (MPGN) and cryoglobulinemia [7]. Chronic kidney disease (CKD) has also been identified as an extrahepatic manifestation of chronic HCV [8] and a factor in poor prognosis for chronic HCV patients [9]. HCV seropositivity accounts for 7.8% of patients with CKD stages ≥ 3 (eGFR < 60 mL/min/1.73 m^2^) in Taiwan, leading to a twofold increase in risk of developing end-stage renal disease (ESRD) [6,10]. In contrast, HBV infection was not associated with CKD in a large-scale observational study [11]. Therefore, the potential interaction between HCV and CKD should not be ignored, and factors related to both HCV and CKD should be emphasized for prevention of disease deterioration, especially in HCV endemic countries.

The pathophysiology of HCV contributing to CKD suggests cirrhotic complications, such as a hepatorenal syndrome, cryoglobulinemia with cryoglobulin deposits in the vasculature [12], and metabolic alterations with insulin resistance or late diabetes [13], which are also predictors of anti-HCV treatment failure. Beyond pre-existing comorbidities, HCV with CKD can progress to ESRD due to unhealthy behaviors including a high-fat and salty diet, inadequate water intake, irregular exercise, or heavy alcohol consumption [10]. While successful viral eradication of HCV promises decreasing incidences of CKD [7,14,15], aggressive promotion of physical activity, healthy diet, and early detection of metabolic syndrome (MetS) might optimize renal function maintenance in this risky population [16,17]. In addition, unhealthy behaviors are associated with MetS, which affect proteinuria and renal atherosclerosis. Furthermore, HCV is a risk factor for MetS, especially dyslipidemia.

Several community-based studies have suggested HCV and CKD are correlated [18,19,20]; however, that research failed to include details of personal habits, and dietary and physical activity. Further, metabolic indices as potential confounders affecting the association of healthy behaviors with HCV in endemic regions are unknown. We hypothesize that maintaining healthy behaviors in HCV patients is related to better prognosis regarding CKD. This study explores the relationship between unhealthy behaviors, metabolic disturbances, and HCV comorbid with CKD.

## 2. Methods

### 2.1. Study Design and Participants

This cross-sectional community-based study was conducted in rural areas on the west coast of southern Taiwan. Health check-ups were performed by a nurse-led community health promotion program between June and December 2019. The inclusion criteria were patients aged 20 to 80 years old who agreed to sign the informed consent form and undergo a series of health examinations, including anthropometric measurements, fundamental laboratory analysis, and serological assays for viral hepatitis. The exclusion criterion was comorbidity of hepatitis B virus (HBV) and HCV. In total, 2439 participants completed a health check-up. Of these, 40 subjects had coinfections of HBV and HCV, and a further 12 individuals had incomplete data. Thus, 2387 participants were eligible for analysis (Figure 1). This study was approved by the Chang Gung Memorial Hospital Institutional Review Board (IRB NO: 201900222A3).

### 2.2. Questionnaires

All research assistants received 2 h of questionnaire interview training by an investigator. Interviews with pilot participants confirmed a 90% rate of inter-rater reliability. All study participants completed a 10 min interview using a structured questionnaire and donated a blood sample after fasting overnight for 8 h. Participants’ demographic and clinical characteristics were collected, including gender, age, education level, dietary habits, exercise, and substance use. “Exercise” was graded based on the answer to, “How often do you take exercise weekly?” as follows: (0) never; (1) 1~2 days/weekly = seldom; (2) over 3 days/weekly = often; (3) every day = always [21]. A patient was classified as irregular exercise if the answer was 0 (never) or 1 (seldom). Healthy dietary habits were defined as: ≥three portions of vegetables per day (one portion is equivalent to 100 g edible vegetables), ≥two portions of fruit per day (one portion is equivalent to 100 g edible fruits), and a water intake ≥ 1500 cc per day, in accordance with the daily dietary guidelines suggested by the Health Promotion Administration of the Taiwan Ministry of Health and Welfare [22].

### 2.3. Anthropometric Measurements and Laboratory Analysis

Clinical examination data related to metabolic syndrome and chronic hepatitis were also obtained. Waist circumference (WC) was measured by a soft tape at the umbilical level while standing, and the mean value of two measurements recorded. Systolic blood pressure (SBP) and diastolic blood pressure (DBP) were measured repeatedly at 5 min intervals in a sitting position after 30 min of resting in a quiet room, using an electronic sphygmomanometer (Omron HEM-1000, MCN0937000 Omron (Dalian) Co., Ltd., Dalian, China). We recorded mean values of both blood pressure measurements. Blood samples of all participants were obtained following an 8 h fast and the following levels collected: high-density lipoprotein (HDL), triglycerides (TG), aspartate aminotransferase (AST), and alanine aminotransferase (ALT) (values were estimated using Roche Diagnostics, Cobas6000, C501, Mannheim, Germany). The estimated glomerular filtration rate (eGFR) was calculated with MDRD GFR equation (GFR in mL/min per 1.73 m^2^ = 175 × SerumCr^−1.154^ × age^−0.203^ × 1.212 (if patient is black) × 0.742 (if female)), and the glycosylated hemoglobin (HbA1c) measured using an autoanalyzer (Trinity Biotech, Premier HB9210, Kansas City, MO, USA).

### 2.4. Serological Assays for Viral Hepatitis

A patient was classified as infected with HCV if the anti-HCV antibody test (SP-NANBASE C-96 3.0 plate, General Biological Corp., Hsinchu, Taiwan) was positive, as infected with HBV if the HBV surface antigen test (General Biological Corp.) was positive, and placed in the non-hepatitis group if both enzyme-linked immunosorbent assay tests were negative. Subjects with HBV and HCV co-infection were excluded from the analysis.

### 2.5. Definition of Metabolic Syndrome

Components related to metabolic syndrome include a WC ≥ 90 cm for males or ≥80 cm for females, SBP > 130 mmHg or DBP > 85 mmHg, HDL < 40 mg/dL for males or <50 mg/dL for females, TG ≥ 150 mg/dL, and HbA1c ≥ 5.7%, according to the International Diabetes Federation (IDF) criteria [23]. Metabolic syndrome was diagnosed if three or more of the five components were present.

## 3. Statistical Analysis

The study subjects’ demographic, dietary, and hepatitis status (non-hepatitis vs. HBV vs. HCV) were compared using a one-way analysis of variance for continuous variables or the chi-square test for categorical variables. For continuous variables with a skewed distribution, the Kruskal–Wallis test was used to compare group differences. Pairwise comparisons were performed using Bonferroni adjustment if the overall difference was significant.

Demographic, dietary, and clinical characteristics of study subjects with and without CKD (defined as eGFR < 60 mL/min/1.73 m^2^) were compared using independent sample *t*-tests for continuous variables or the chi-square test for categorical variables. For continuous variables with a skewed distribution, the Mann–Whitney U-test was used to compare groups. A series of univariate logistic regression analyses were conducted to screen for factors potentially associated with CKD. Variables with a significance level less than 0.15 were then incorporated into a multivariable logistic regression model with backward elimination [24]. All tests were two-tailed, and *p* < 0.05 was considered statistically significant. Data analyses were conducted using IBM SPSS Statistics for Windows, version 25 (I.B.M. Corp., Armonk, NY, USA).

## 4. Results

### 4.1. Characteristics of Participants

Of the 2387 participants in this study, 306 (12.8%) and 213 (8.9%) were seropositive for HCV and HBV, respectively (Table 1). The mean age of participants was 64.1 years old (y/o), and the majority were female (63.2%). The HCV group was older (70.8 ± 10.0 y/o), had a higher female proportion (68.6%), a lower education level, more unhealthy dietary habits, and greater betel consumption compared to the non-HCV groups. The HCV group also had higher SBP, HbA1c, TG, and lower HDL levels (*p* < 0.05). MetS occurred in significantly more HCV seropositive subjects (prevalence of MetS in HCV, HBV, and the non-hepatitis group was 58.8%, 41.8%, and 52.1%, respectively; *p* < 0.001). Higher mean AST and ALT and lower mean eGFR were also observed in the HCV group. A significantly higher proportion of HCV seropositive participants had eGFR < 60 mL/min/1.73 m^2^ (23.9% vs. 8.9% in HBV, and 13.4% in the non-hepatitis group). No effects of gender, WC, exercise, smoking, or alcohol consumption were observed among the three groups (Table 1).

### 4.2. Differences between CKD and Non-CKD Groups

We next compared the characteristics of subjects with and without CKD (Table 2). The mean eGFR was 46.9 ± 11.5 and 90.9 ± 19.8 mL/min/1.73 m^2^ in CKD and non-CKD groups, respectively. Individuals with CKD were older, less likely to be female, had a lower education level, had poorer dietary habits, and were less likely to exercise regularly. Subjects with CKD also had a higher frequency of components of metabolic syndrome, including higher WC, SBP, HbA1c, and TG, lower DBP, and lower HDL. The prevalence of metabolic syndrome was higher in the CKD group than in the non-CKD group (68.1% vs. 49.3%, *p* < 0.001). There were no differences in substance use, including smoking, betel use, and alcohol use, between the two groups. The CKD group had poorer liver function (ALT and AST). The prevalence of CKD was highest in the HCV group (23.9%), followed by the non-hepatitis group and HBV group (13.4% and 8.9%, respectively; *p* < 0.001; Table 2).

### 4.3. Potential Risks Associated with CKD

All variables were included in a series of univariate logistic regression analyses to investigate factors potentially associated with CKD (Table 3). In the multivariable analysis, the following variables were retained: older age (odds ratio (OR) 1.08, 95% confidence interval (CI) 1.07–1.10), less frequent exercise (OR 1.38, 95% CI 1.04–1.83), higher WC (OR 1.02, 95% CI 1.01–1.04), lower HDL (OR 0.97, 95% CI 0.96–0.98), higher HbA1c (OR 1.14, 95% CI 1.02–1.26), and the presence of HCV (OR 1.44, 95% CI 1.05–1.98) were independently associated with a higher risk of CKD.

### 4.4. The Relationship between MetS Components and eGFR

As shown in Figure 2, a greater reduction in renal function was associated with a higher number of MetS components in HCV seropositive subjects (one, three, and five components of MetS associated with a median eGFR of 86.4, 77.1, and 64.5 mL/min/1.73 m^2^, respectively). The effect of irregular exercise on CKD risk was analyzed in the groups stratified by hepatitis status. The effect of irregular exercise on CKD risk was similar among the non-hepatitis, HBV, and HCV groups (Table 4).

## 5. Discussion

This community-based study investigated residents in an HCV endemic area of Taiwan and found that HCV seropositivity rate was higher than the overall rate in Taiwan (12.8% vs. 2–4%) [1,18]. In addition, the HCV group had a higher proportion of individuals with CKD compared to both the HBV group and the non-hepatitis group. HCV seropositivity was significantly related to CKD; risk of CKD increased by 44% compared to non-hepatitis participants (OR 1.44, 95% CI 1.05–1.98). This result matches those of prior studies [3,4,5], including a previous meta-analysis (HR 1.45 in Asia, 95% CI 1.27–1.65) [25]. Lee et al. [18] also found that HCV is significantly associated with advanced CKD, with a higher prevalence of HCV seropositivity (14.5%) in later CKD stages. Therefore, early detection of HCV and reduction in associated risk factors might offer benefits for preventive care in the CKD population.

Previous research suggests effective viral eradication for HCV decreases risk of incidental CKD by 30% [7]. However, therapeutic failure of anti-HCV therapy is likely in the presence of other risk factors, such as large waist circumference, diabetes, and old age [13]. In our study, age was associated with a higher prevalence of both HCV seropositivity and CKD, in agreement with previous investigations [10]. A failure to recognize disease progression during asymptomatic stages of CKD in HCV patients may lead to a failure to initiate prompt therapy. Similarly, evidence indicates that early detection and treatment of HCV infection in CKD patients is crucial to preserving renal function and preventing deterioration to ESRD. Thus, regular and continuous monitoring of renal function in vulnerable HCV patients should be conducted and results shared with patients to encourage cooperation with strict treatment. However, the renal benefit of antiviral therapy might be offset if unhealthy behaviors and MetS burden coexist frequently in HCV and CKD patients. Therefore, comprehensive lifestyle, metabolic indices, and viral hepatitis surveys would be beneficial in HCV endemic countries [16,17].

The proportion of MetS (58.8%) in the HCV seropositive population is higher than that seen in other HCV cohort studies (24.7–35%) [26]. The differences are likely a result of our study population coming from a rural area that was older and had a different lifestyle than an urban population. HCV patients with metabolic disturbances have a more rapid progression of liver fibrosis and hepatocellular carcinoma [27]. Studies have reported that MetS significantly increases the risk of CKD (using the IDF criteria) by 32–50% and is associated with a rapid decline in eGFR [17,28,29,30]. HCV seropositive subjects in our study shared similar features to patients with CKD, including an older age, higher HbA1c, and lower HDL, which contribute to advanced diabetic kidney disease [31]. Our study shows an escalating response of decline in eGFR with increasing components of MetS (presence of one vs. five components of MetS were associated with a median eGFR of 86.4 and 64.5 mL/min/1.73 m^2^, respectively). Individuals with HCV and MetS were more likely to fail antiviral treatment [27], and incomplete viral eradication can lead to CKD due to extrahepatic immunity aggregation. Accordingly, the coexistence of MetS and HCV is critical for CKD; failure to control either condition will lead to renal function declines with subsequent advanced CKD, worsening to ESRD. Prompt measures to control components of MetS might delay the deterioration process. Implementing healthy behaviors and dietary habits could decrease MetS burden and renal function dysfunction, especially for those HCV seropositive participants with unhealthy lifestyle habits.

The effects of exercise on eGFR and lipid alternation in current studies are conflicting. In one study, regular exercise improved physical performance and fitness, cardiopulmonary function, and quality of life in patients with CKD and led to improved eGFR and decreased BMI [16]. Other studies suggest that exercise-improved HDL-C levels did not significantly change eGFR [32,33,34]. Our study shows that a lack of regular exercise is an independent risk factor for CKD, regardless of the presence of HCV. We also assessed the effect of irregular exercise in patient groups stratified by viral seropositivity and found that physical activity may ameliorate CKD in individuals with HCV who are older and relatively less-educated. Considering the average rate of irregular exercise is extremely high (average 69.5%) in this rural area, community health efforts to enhance exercise education and promote regular physical activity can improve quality of life and preserve renal function in HCV patients with CKD.

In our study, subjects with HCV or CKD seemed to have predominately unhealthy dietary behaviors, eating fewer vegetables and fruit, and drinking less water per day. Dietary interventions have been proposed to improve CKD outcomes in the last decade; maintaining healthy dietary behaviors reduces CKD risk and slows deterioration in renal function. A high-protein diet was associated with a 1.32-fold-increased risk of rapid decline in eGFR [35] and a higher risk of advanced liver fibrosis, while soy supplements were related to lower liver inflammation [36]. However, unhealthy dietary behaviors lacked significant influence on CKD in our investigation.

## 6. Limitations

This study has several limitations. First, the cross-sectional study lacked sequential and long-term observations of renal function (eGFR), which limited estimates on the speed of renal deterioration in CKD comorbid with HCV or MetS. In addition, many factors associated with eGFR were not detected comprehensively in the community-based investigation, including uric acid level and nephrotoxic medication use. Patients with comorbidities such as heart failure and liver cirrhosis were not excluded. Second, we could not obtain detailed data on viral activity or viral load in annual health check-ups, and the effect of acute hepatitis on CKD may differ from that of chronic infection and recovered hepatitis. Patients receiving antiviral therapy did not undergo subgroup analysis, as renal function results of HCV patients that received viral eradiation therapy were unavailable.

A third limitation is that the data came from a rural area with high levels of HCV, in which participants also tended to be elderly in age and predominately female. These phenotypes may not be similar to the presentation worldwide or nationwide, and phenotype stratification was beyond the assessments in this study, limiting the application of study results to other regions. This study also excluded HCV and HBV combined seropositive patients, which might overestimate or underestimate study results. Fourth, the study was based on data from a community health promotion program conducted in a local hospital during the daytime. Thus, patients with fixed working hours or limited ambulatory ability may not have participated in this program, leading to the exclusion of more male individuals and more severely ill patients. Furthermore, we could not obtain data on the intensity of physical activity and exercise, and variation of effect on CKD is unclear. Finally, as in other studies containing questionnaire surveys, response bias could exist.

## 7. Conclusions

HCV seropositivity, metabolic disturbances, and irregular exercise were independently associated with CKD. Based on this association, we suggest that aggressive health promotion of physical activity might reduce the occurrence and deterioration of CKD in HCV. However, large-scale cohort studies are needed to explore further the associations between HCV, MetS, and renal dysfunction.

## Figures and Tables

**Figure 1 ijerph-19-03558-f001:**
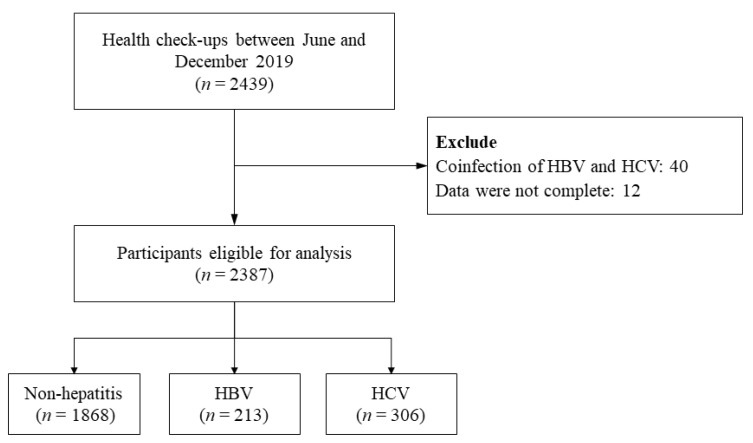
Flowchart of the participant enrollment.

**Figure 2 ijerph-19-03558-f002:**
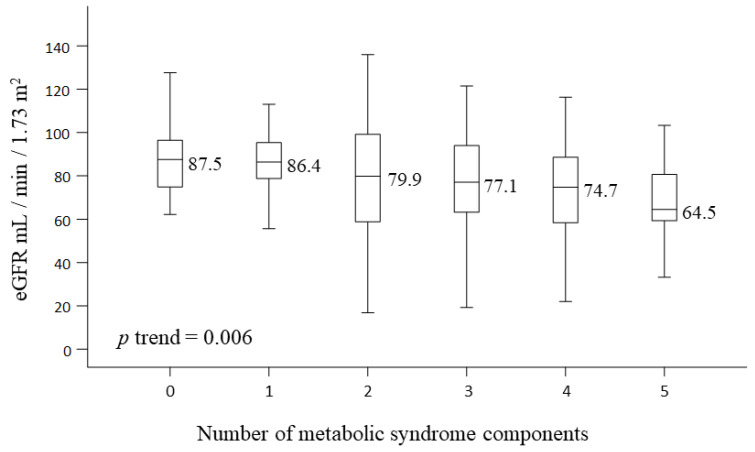
Renal function (eGFR) with the number of different metabolic syndrome components present in subjects with hepatitis C virus infection. The *p*-value for trend was obtained from the linear contrast in the general linear model. eGFR, estimated Glomerular filtration rate.

**Table 1 ijerph-19-03558-t001:** Demographics and characteristics of the study subjects according to the HBV and HCV status (*N* = 2387).

Variable	Total	Non-Hepatitis	HBV	HCV	*p*
Number of subjects	2387	1868	213	306	
Female	1508 (63.2)	1160 (62.1)	138 (64.8)	210 (68.6)	0.079
Age, years	64.1 ± 14.9	63.3 ± 15.6	61.1 ± 12.3	70.8 ± 10.0	<0.001
Age group					<0.001
<40 years	213 (8.9)	199 (10.7)	14 (6.6)	0 (0.0)	
40–64 years	838 (35.1)	653 (35.0)	104 (48.8)	81 (26.5)	
≥65 years	1336 (56.0)	1016 (54.4)	95 (44.6)	225 (73.5)	
Education level, years	6.0 (0.0, 12.0)	6.0 (0.0, 12.0)	6.0 (6.0, 12.0)	6.0 (0.0, 6.0)	<0.001
Dietary habits					
Intake vegetable ≥ 3 portions per day	1592 (66.7)	1279 (68.5)	140 (65.7)	173 (56.5)	<0.001
Intake fruit ≥ 2 portions per day	1337 (56.0)	1079 (57.8)	118 (55.4)	140 (45.8)	<0.001
Intake of water ≥ 1500 cc per day	1402 (58.7)	1139 (61.0)	118 (55.4)	145 (47.4)	<0.001
Irregular exercise	1660 (69.5)	1295 (69.3)	145 (68.1)	220 (71.9)	0.589
Substance use					
Smoking	427 (17.9)	332 (17.8)	37 (17.4)	58 (19.0)	0.864
Betel	221 (9.3)	164 (8.8)	14 (6.6)	43 (14.1)	0.005
Alcoholic drinking	241 (10.1)	183 (9.8)	29 (13.6)	29 (9.5)	0.200
Data of metabolic syndrome (MetS)					
Waist circumference (WC), cm	84.8 ± 10.8	84.8 ± 10.9	84.3 ± 10.7	85.2 ± 10.0	0.611
Systolic blood pressure, mmHg	134.7 ± 20.4	134.7 ± 20.1	131.6 ± 21.0	137.0 ± 21.2	0.013
Diastolic blood pressure, mmHg	81.7 ± 12.4	81.8 ± 12.2	82.3 ± 13.1	80.4 ± 12.6	0.141
High-density lipoprotein, mg/dL	51.0 ± 13.2	51.2 ± 13.0	52.9 ± 13.5	49.1 ± 13.9	0.004
Glycosylated hemoglobin, mg/dL	6.1 ± 1.1	6.1 ± 1.0	6.0 ± 1.0	6.3 ± 1.3	0.011
Triglyceride, mg/dL	113 (0, 166)	114.0 (81, 170)	93 (69, 139)	118 (83, 165)	<0.001
MetS	1242 (52.0)	973 (52.1)	89 (41.8)	180 (58.8)	0.001
Liver and renal function					
ALT, U/L	19.0 (14.0, 27.0)	18.5 (14.0, 26.0)	21.0 (16.0, 29.0)	19.5 (14.0, 29.0)	<0.001
AST, U/L	23.0 (19.0, 28.0)	23.0 (19.0, 27.0)	24.0 (20.0, 30.0)	25.0 (20.0, 31.0)	<0.001
eGFR, mL/min/1.73 m^2^	84.6 ± 24.4	85.4 ± 24.5	87.8 ± 21.7	77.2 ± 23.7	<0.001
eGFR < 60 mL/min/1.73 m^2^	342 (14.3)	250 (13.4)	19 (8.9)	73 (23.9)	<0.001

HBV, hepatitis B virus; HCV, hepatitis C virus; ALT, alanine aminotransferase; AST, aspartate aminotransferase; eGFR, estimated Glomerular filtration rate; Data are presented as mean ± standard deviation, frequency, and percentage or median (25th, 75th percentile).

**Table 2 ijerph-19-03558-t002:** Demographics and characteristics of the study subjects according to the renal function status (*N* = 2387).

Variable	eGFR < 60 mL/min/1.73 m^2^	eGFR ≥ 60 mL/min/1.73 m^2^	*p*
Number of subjects	342	2045	
Female	194 (56.7)	1314 (64.3)	0.008
Age, years	74.8 ± 9.1	62.3 ± 15.0	<0.001
Age group			<0.001
<40 years	0 (0.0)	213 (10.4)	
40–64 years	38 (11.1)	800 (39.1)	
≥65 years	304 (88.9)	1032 (50.5)	
Education level, years	0.0 (0.0, 6.0)	6.0 (0.0, 12.0)	<0.001
Dietary habits			
Intake vegetable ≥ 3 portions per day	196 (57.3)	1396 (68.3)	<0.001
Intake fruit ≥ 2 portions per day	149 (43.6)	1188 (58.1)	<0.001
Intake of water ≥ 1500 cc per day	172 (50.3)	1230 (60.1)	0.001
Irregular exercise	257 (75.1)	1403 (68.6)	0.015
Substance use			
Smoking	71 (20.8)	356 (17.4)	0.134
Betel	41 (12.0)	180 (8.8)	0.060
Alcoholic drinking	38 (11.1)	203 (9.9)	0.501
Data of metabolic syndrome (MetS)			
Waist circumference (WC), cm	88.3 ± 10.1	84.2 ± 10.8	<0.001
Systolic blood pressure, mmHg	137.5 ± 21.7	134.2 ± 20.1	0.006
Diastolic blood pressure, mmHg	79.2 ± 13.4	82.1 ± 12.1	<0.001
High-density lipoprotein, mg/dL	45.2 ± 12.7	52.0 ± 13.0	<0.001
Glycosylated hemoglobin, mg/dL	6.4 ± 1.2	6.0 ± 1.0	<0.001
Triglyceride, mg/dL	134.0 (99.0, 194.0)	110.0 (77.0, 162.0)	<0.001
MetS	233 (68.1)	1009 (49.3)	<0.001
Liver and renal function			
ALT, U/L	18.0 (13.0, 25.0)	19.0 (14.0, 27.0)	0.013
AST, U/L	24.0 (19.0, 31.0)	23.0 (19.0, 28.0)	0.007
eGFR, mL/min/1.73 m^2^	46.9 ± 11.5	90.9 ± 19.8	<0.001
HBV and HCV status			<0.001
Non-hepatitis	250 (13.4)	1618 (86.6)	
HBV	19 (8.9)	194 (91.1)	
HCV	73 (23.9)	233 (76.1)	

eGFR, estimated Glomerular filtration rate; ALT, alanine aminotransferase; AST, aspartate aminotransferase; HBV, hepatitis B virus; HCV, hepatitis C virus; Data are presented as mean ± standard deviation, frequency, and percentage or median (25th, 75th percentile).

**Table 3 ijerph-19-03558-t003:** Association between demographics, characteristics, and the risk of chronic kidney disease of the study subjects (*N* = 2387).

Explanatory Variable	Univariate Analysis		Multivariable Analysis	
	OR (95% CI)	*p*	OR (95% CI)	*p*
Female	0.73 (0.58–0.92)	0.008		
Age, per year	1.09 (1.08–1.10)	<0.001	1.08 (1.07–1.10)	<0.001
Education level, per year	0.88 (0.86–0.90)	<0.001		
Intake vegetable ≥ 3 portions per day	0.62 (0.49–0.79)	<0.001		
Intake fruit ≥ 2 portions per day	0.56 (0.44–0.70)	<0.001	0.79 (0.61–1.01)	0.065
Intake of water ≥ 1500 cc per day	0.67 (0.53–0.84)	0.001		
Irregular exercise	1.38 (1.06–1.80)	0.015	1.38 (1.04–1.83)	0.027
Smoking	1.24 (0.93–1.65)	0.135		
Betel	1.41 (0.98–2.02)	0.061		
Alcoholic drinking	1.13 (0.79–1.64)	0.501		
Waist circumference, cm	1.04 (1.03–1.05)	<0.001	1.02 (1.01–1.04)	0.001
Systolic blood pressure, mmHg	1.01 (1.002–1.013)	0.006		
Diastolic blood pressure, mmHg *	0.98 (0.97–0.99)	<0.001		
High-density lipoprotein, mg/dL	0.96 (0.95–0.97)	<0.001	0.97 (0.96–0.98)	<0.001
Glycosylated hemoglobin, mg/dL	1.31 (1.20–1.43)	<0.001	1.14 (1.02–1.26)	0.015
Triglyceride, mg/dL	1.002 (1.001–1.003)	<0.001		
ALT, U/L	1.01 (0.99997–1.015)	0.051		
AST, U/L #	0.99 (0.988–1.002)	0.146		
HBV and HCV status				
Non-hepatitis	Reference	Reference	Reference	Reference
HBV	0.63 (0.39–1.03)	0.068	0.90 (0.53–1.51)	0.679
HCV	2.03 (1.51–2.72)	<0.001	1.44 (1.05–1.98)	0.025

OR, odds ratio; CI, confidence interval; BMI, body mass index; ALT, alanine aminotransferase; AST, aspartate aminotransferase; eGFR, estimated Glomerular filtration rate; HBV, hepatitis B virus; HCV, hepatitis C virus; * Diastolic blood pressure was not included in the multivariable model due to its collinearity with systolic blood pressure; # AST was not included in the multivariable model due to its collinearity with ALT.

**Table 4 ijerph-19-03558-t004:** The risk of renal impairment in subjects with and without regular exercise stratified by the HBV and HCV status.

Subgroup	Number of Chronic Kidney Disease (%)		OR (95% CI) of Irregular Exercise	*p* for Interaction
	Irregular Exercise	Adopt Regular Exercise		
HBV and HCV status				0.503
Non HBV and Non HCV	185 (14.3)	65 (11.3)	1.30 (0.96–1.76)	
HBV only	13 (9.0)	6 (8.8)	1.02 (0.37–2.80)	
HCV only	59 (26.8)	14 (16.3)	1.88 (0.99–3.59)	

HBV, hepatitis B virus; HCV, hepatitis C virus; OR, odds ratio; CI, confidence interval.

## Data Availability

The individual-level data used and/or analyzed for the current study are available from the corresponding author on request.

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
