# Peer review of "The Impact of Hepatitis C Virus, Metabolic Disturbance, and Unhealthy Behavior on Chronic Kidney Disease: A Secondary Cross-Sectional Analysis"

_ijerph, 2022, doi:10.3390/ijerph19063558_

Round 1

Reviewer 1 Report

You present an interesting topic and one that is actively researched globally. There are interesting associations presented, but overall the study lacks significant novelty. More so, there are significant limitations that may make the findings unreliable. Substantial work has to be done on proofreading. Overall, comments and suggestions are as follows. 

Abstract:

-"This study investigated the relationships between HCV and CKD under-considered cofounders as unhealthy behaviors and metabolic disturbances" - this sentence structure is not well built and difficult to follow. Something like this may be more appropriate: "This study investigated the relationships between HCV and CKD, to include under considered cofounders such as unhealthy behaviors and metabolic disturbances"

HCV  emerged  to  significantly  increase 44% risk of CKD compared to non-HCV  - again sentence structure is not well built. Would be better to express as HCV emerged to significantly increase the risk of CKD compared to non-HCV by 44% or something similar as per your discretion 

Summary Box - why are you using past perfect tense? In second paragraph, "irregular  physical  exercise  could  be  the  most  one  of  un-healthy lifestyles" - again poor sentence build - needs proof-reading. 

Introduction:

Again, would avoid discussing what is known so far and what has been studied in the past and past perfect tense - so to say, becomes less "palatable" to read. As an example "Chronic HCV infection would lead to multiple hepatic complications, including advanced liver cirrhosis, hepatocellular carcinoma,  and  worsening renal  function[3-5],  and  was  related  to two-fold  higher  all-cause mortality[6]" - would be better off as "Chronic HCV infection is known to lead to multiple hepatic complications..., and is associated with a two-fold higher risk for all-cause mortality"

Overall as the introduction continues, there are multiple areas that would require proof-reading and correction of the sentence structures. Valid and sound ideas are presented, but again, needs proofreading. 

Methods:

2.1 

would be prudent to mention exclusion criteria here instead of leaving to the reader to find out that co-infection with HBV was excluded. More so, sentence structure would need to be changed. As an example, "Totally 2,439 participants completed the health check-up. Of whom, 40 subjects had coinfection of HBV and HCV, and data of 12 individuals were not complete" - this may read better as "In total, 2439 participants completed the health check-up, of whom 40 subjects were excluded due to coinfection of HBV and HCV, while 12 more subjects had incomplete data"

2.2

Would start the sentence as "healthy dietary habits were defined as...." and then going through the various criteria 

2.3

"during index of examination" - this does not sound right, needs proofreading as above 

"The estimated glomerular  filtration rate  (eGFR) was calculated carefully" - carefully is not that important, but rather what formula you used and why with a proper reference 

2.4

-why would one define HCV infection as positive serology? HCV RNA was not checked? Only serologic positivity does not define ongoing infection and hence, requires no further evaluation or treatment 

Results:

4.1

"while 306 (12.8%) and 213 (8.9%) of all were seropositive for HCV and HBV, respectively" - I would suggest to remove "of all"

need to follow the same tense. If started in the past tense, need to follow with the past tense the subsequent sentences when describing the results. 

would also recommend to concentrate on the main results, rather than listing all the findings that were statistically significant, as these findings are also present in the table and reader can use that as a reference 

4.3 - title does not sounds appealing - would not start a title with "those potential risks"

Discussion

"This  community-based  study  investigated residents  of  an HCV  endemic area, in which12.8% of all participants were HCV seropositive, revealed a higher percentage than the overall prevalence in Taiwan ranged from 2% to 4%" - this sentence again requires proofreading and adjustments to make it easier to read and understand. As an example "this community-based study investigated residents of an HCV endemic area, and found a much higher prevalence of HCV seropositivity than the overall prevalence in Taiwan (12.8% vs 2 - 4%). 

To support the argument that in your study it was indeed HCV that was associated with CKD, I would have liked to see the data of multivariate regression for confounders such as age, as CKD is more prevalent with age and renal function normally deteriorates with age, while your HCV positive population was clearly older than the non-HCV group. More so, as mentioned previously, knowing HCV RNA status would have shed more light as to whether HCV was indeed contributing to the higher risk. 

The second paragraph requires much proofreading as as above, without knowing the HCV RNA status of your patients, it is even impossible to talk about the benefits of eradication therapy

The subsequent paragraph again requires much proofreading but ideas are better presented and follow logical thought process. The same for the next paragraphs. 

Limitations

You mentioned about HCV RNA not available as your limitation, but it is indeed a huge limitation when talking about HCV infection. Otherwise, apart from proofreading needs, well done with presenting the limitations. 

Conclusions:

"Effective viral eradication therapy and aggressive health promotion  of  physical  activity might reduce  the  occurrence and  deterioration of  CKD" - I think this is a rather bold statement based on the evidence presented in your study. Firstly, as mentioned above, it is unknown whether your subjects had any active or chronic HCV infection, and their infections could have been long eradicated. Second, even though the above statement is true from other available evidence, this is not as strongly supported by your study findings. 

Author Response

Dear reviewer 1. Thank you for considering our manuscript “The impact of hepatitis C virus, metabolic disturbance, and unhealthy behavior on chronic kidney disease: A secondary cross sectional analysis”. We sincerely appreciate the comments from all reviewers, and wish the revision meet your requirements. The manuscript has been proofread to make it easier to read and understand. Because the entire manuscript has been edited in English to correct the sentence structure, details of proofreading and words adjustments are not explained one by one. The alterations made to clarify the research concepts and arguments based on your opinions were explained and listed below.
2.1. Study design and participants Exclusion criteria is explained, and sentence structures are corrected by” The exclusion criterion was comorbidity of hepatitis B virus (HBV) and HCV. In total, 2,439 participants completed a health check-up. Of these, 40 subjects had coinfections of HBV and HCV, and a further 12 individuals had incomplete data.”
2.3. Anthropometric measurements and Laboratory analysis The equation of GFR used in our study (MDRD GFR equation) was listed in the text.
2.4. Serological assays for viral hepatitis
A positive anti-HCV antibody test represents acute or chronic hepatitis infection, recovered hepatitis, occult infection, or chronic carrier state. The study aimed to understand the effects of exposure to HCV on CKD. Although effects on CKD might differ on different infection status, the current study design had made it difficult to clarify the relationship. This cross-sectional community-based study was based on health check-ups in a community health promotion program, the anti-HCV antibody test was used as a screen tool to detect asymptomatic infected patients. Patients with positive result were informed and suggested to consult a physician for further surveys. However, some of the patients with positive anti-HCV antibody test may not visit a doctor, and whether HCV RNA should be tested depends on the physician's clinical assessment. Furthermore, selection bias could exist in the analysis based on patients
who had HCV RNA tested. The relationship between different infection status of
HCV and CKD needs further investigations.
Discussion
As you had mentioned, HCV positive population was clearly older than the
non-HCV group, the result had made it difficult to interpret the association between
HCV and CKD. Therefore, multivariable analysis was carried out in order to
investigate confounders in CKD, the result on Table 3 had revealed that both old age
(OR 1.08, 95% CI 1.07–1.10) and HCV (OR 1.44, 95% CI 105.–1.98) were
independently associated with a higher risk of CKD.
Limitation
More details were revealed to understand possible bias according to reviewers’
suggestions. We edit the sentence to “the effect of acute hepatitis on CKD may differ
from that of chronic infection and recovered hepatitis.”
Conclusion
We rephrased the conclusion, and deleted “effective viral eradication therapy” to
avoid misinterpretations and focus on the main findings.

Reviewer 2 Report

From my point of view, it is a well-structured manuscript, with a well-planned work.
As its authors comment on the limitations of the study, it would be interesting to carry out a prospective study to see if lifestyle modification can reduce or prevent the progression of CKD.
Regarding the risk factors that have been taken into account, I miss including uric acid (since high levels are associated with greater progression of CKD) and knowing if patients chronically consume any potentially nephrotoxic drug (NSAIDs , cancer treatments ...)

Reviewer 3 Report

In the present manuscript by Wang et al. titled » The impact of hepatitis C virus, metabolic disturbance, and unhealthy behavior in chronic kidney disease: a secondary cross-sectional analysis« the authors aim to explore the relationships between unhealthy behaviors, metabolic disturbances, and HCV with CKD. The study represents a cross-sectional community-based study of patients between 20 and 80 years carried out in Taiwan. The manuscript represents an interesting study from a HCV endemic area, which brings forward further insight into the effects of HCV on kidney health. Below are some suggestions to improve the manuscript.

  • The language is generally understandable, but the manuscript could benefit from having language editing.
  • The methods section contains several descriptions on how the study was performed. Additionally, the authors should report how exactly was the blood pressure measured, i.e. which arm was used, were the patients sitting or lying down, were they resting in a quiet room, had they not drunk caffeine before etc.
  • Furthermore, the authors mention that the eGFR was carefully calculated but do not report which equation was used. Currently the CKD-EPI 2021 creatinine equation is the newest and most precise method for the population, which can be further expanded to use cystatin C if available. This is important as the mean eGFR was quite high and other equations (e.g. MDRD) have shown inferior precision. A small comment is that the statistical analysis section is marked 2 (should be 3) and that the authors should move the definition of CKD when it is first mentioned (“… subjects with and without CKD were compared …”)
  • Anti-HCV antibody test and the HBs antigen test were used for hepatitis definition, which would show acute or chronic hepatitis infection, however the authors do not mention possibility of recovered hepatitis, occult infection, or chronic carrier state. They shortly mention this in the limitations but could rephrase hepatitis into “active hepatitis” in other parts of the manuscript.
  • The authors should specifically mention other cofounders that could affect eGFR such as medication (ACEI, SGLT2, NSAIDs, etc.), history of kidney disease (IgA nephropathy, PCKD, urinary tract obstruction, etc.), presence of comorbidities (cirrhosis, heart failure, ischemic heart disease) in the limitations of the study if this data is not available.
  • In the conclusion the authors that: “Effective viral eradication therapy…” might reduce deterioration of CKD, however the study was not designed to measure this, which is why they should rephrase this.

Considering all comments, I think the article should have several minor revisions before being considered for publication.

Round 2

Reviewer 1 Report

Thank you for the corrections and following the suggestions. 

This manuscript is a resubmission of an earlier submission. The following is a list of the peer review reports and author responses from that submission.